# Median Arcuate Ligament Compression in Orthotopic Liver Transplantation: Results from a Single-Center Analysis and a European Survey Study

**DOI:** 10.3390/jcm8040550

**Published:** 2019-04-23

**Authors:** Zoltan Czigany, Joerg Boecker, Daniel Antonio Morales Santana, Jan Bednarsch, Franziska Alexandra Meister, Iakovos Amygdalos, Peter Isfort, Martin Liebl, Ulf Peter Neumann, Georg Lurje

**Affiliations:** 1Department of Surgery and Transplantation, University Hospital RWTH Aachen, 52074 Aachen, Germany; zczigany@ukaachen.de (Z.C.); jboecker@ukaachen.de (J.B.); dmoralessant@ukaachen.de (D.A.M.S.); jbednarsch@ukaachen.de (J.B.); fmeister@ukaachen.de (F.A.M.); iamygdalos@ukaachen.de (I.A.); uneumann@ukaachen.de (U.P.N.); 2Institute of Radiology, University Hospital RWTH Aachen, 52074 Aachen, Germany; pisfort@ukaachen.de (P.I.); mliebl@ukaachen.de (M.L.); 3Department of Surgery, Maastricht University Medical Centers (MUMC), 6202 AZ Maastricht, The Netherlands

**Keywords:** orthotopic liver transplantation, Dunbar syndrome, median arcuate ligament syndrome, MALS, surgical complications, clinical outcome

## Abstract

Median arcuate ligament compression (MALC) potentially causes arterial complications in orthotopic liver transplantation (OLT). Here we aimed to investigate the incidence of MALC and its impact on clinical outcome after OLT. In addition, we performed an international survey among 52 European liver transplant centers to explore local protocols on the management of these patients. Data of 286 consecutive OLT recipients from a prospective database were analyzed retrospectively (05/2010-07/2017). Preoperative computed-tomography images were evaluated. Celiac axis stenosis due to MALC was found in 34 patients (12%). Intrinsic stenosis was present in 16 (6%) patients. Twenty-six patients (77%) with MALC underwent standard arterial revascularization with median arcuate ligament (MAL)-division. Patients treated for MALC had comparable baseline data and no difference was found in early- and long-term outcome compared to the rest of our cohort. Our survey found heterogeneous strategies regarding diagnosis and treatment of MALC. Only 29% of the centers reported the division of MAL in these patients as routine procedure. Even though there is no consensus on diagnosis and management of MALC among European centers, a surgical division of MAL is feasible and safe and should be considered in OLT recipients with MALC.

## 1. Introduction

Orthotopic liver transplantation (OLT) has evolved as the standard treatment for end-stage liver disease [1]. Despite standardization of the operative technique, hepatic artery thrombosis (HAT) remains a rare but dreadful complication following OLT [2,3,4,5,6].

The vascular supply of the upper abdominal viscera arises mostly from the celiac axis. Median arcuate ligament compression (MALC) is a rare medical condition caused by the extrinsic compression of the celiac axis by means of the fibrous bands of the median arcuate ligament (MAL) and periaortic ganglionic tissue [7,8]. This anatomical condition is usually asymptomatic due to sufficient collateral blood supply but may present with functional ischemia in 10% to 24% [8,9]. Typical clinical presentations of MALC, defined as Dunbar or median arcuate ligament syndrome (MALS), include non-specific abdominal pain (94%), postprandial abdominal pain (80%), weight loss (50%), bloating (39%), nausea and vomiting (56%), and abdominal pain triggered by exercise (8%) [8]. While MALC was first described by Lipshutz et al. more than hundred years ago [10], Harjola and Dunbar reported on the first successful surgical divisions in 1963 and 1965 respectively [11,12].

Abnormalities of the celiac axis and their correlation with clinical outcomes have been demonstrated in patients undergoing curative resection for oesophageal- and pancreatic cancer [7,13]. Even though most of the patients are asymptomatic in the preoperative phase, the lack of sufficient abdominal collaterals and a compromised blood flow to the corresponding tissues postoperatively is likely to be the major cause of morbidity in these patients [7]. In a recent large cohort of 481 oesophagectomies reported by Lainas et al., celiac artery stenosis was associated with oesophageal conduit necrosis following Ivor Lewis procedure, of which 50% was due to MALC [7]. In OLT only scarce clinical evidence is available with regards to the incidence, clinical relevance, diagnosis, and treatment of MALC [14,15,16,17,18,19,20,21].

The present study aimed to analyze the incidence of vascular abnormalities with a special focus on MALC and celiac axis stenosis and the feasibility/safety of MAL division as well as its association with short- and long-term outcome after OLT. Furthermore, an international cross-sectional survey was conducted among 52 European liver transplant centers to provide a comprehensive overview of the current clinical approach in diagnostics and treatment of MALC in OLT in Europe.

## 2. Patients and Methods

### 2.1. Patients and Ethics

Between May 2010 and July 2017, all consecutive patients undergoing OLT at the University Hospital RWTH Aachen (UH-RWTH), Aachen, Germany were considered for inclusion in the present study. Patients with no sufficient pre-transplant imaging were excluded. Furthermore, patients undergoing simultaneous liver-kidney transplantation, those having a living donor liver transplantation, and pediatric liver transplants (recipient age <18) were excluded. Patients undergoing re-OLT were assessed related to the primary transplantation and sub-sequential transplantations were included within the follow-up. The analysis aimed to determine the prevalence of vascular alterations with special focus on celiac axis stenosis due to MALC and the feasibility/safety of MAL division as well as its association with short- and long-term outcome following OLT. The study was conducted at the UH-RWTH in accordance with the current version of the Declaration of Helsinki as well as the Declaration of Istanbul, and good clinical practice guidelines. The study was approved by the responsible Institutional Review Board of the RWTH-Aachen University (EK 047/18). Informed consent was waived due to the retrospective study design and collection of readily available clinical data.

### 2.2. Data Collection and Follow-Up

Data were retrieved from a prospective database and analyzed retrospectively. Baseline donor and recipient characteristics were collected. Various OLT risk-scores were calculated as described before [22]. Postoperative morbidity and mortality were assessed within the first 90-days post-OLT and classified according to the Clavien–Dindo classification and the Comprehensive Complication Index (CCI) [23,24]. Extended criteria donor (ECD) allografts were defined according to the definitions of the German Medical Chamber [25,26]. The Olthoff criteria were adopted to assess post-transplant early allograft dysfunction (EAD) [27]. Early major arterial complications were defined as: complication of the reconstructed hepatic artery threatening with graft loss or death and requiring emergency interventional or surgical treatment during the first 90-days. Postoperative transfusions were defined, as blood products given within the first 7-days following OLT. Blood products administered later during the first 90-days but after 7-days were assessed within the postoperative complications according to the recommendations of the Clavien–Dindo classification [23]. ICU-stay represents the initial stay after the OLT procedure until the transfer of the patient to the standard care HPB/transplantation ward. ICU readmission was assessed within the total hospital stay. Hospital stay was defined by the day of discharge from the UH-RWTH. Our transplantation outpatient clinic as well as the responsible general practitioner/hepatologist provided all follow-up data assessed in this study.

### 2.3. Surgical- and Perioperative Approach

Listing indications were discussed within a multidisciplinary liver transplantation board and followed the German national and Eurotransplant guidelines. Organ procurement and allocation followed the national and Eurotransplant regulations. No organs were procured from prisoners.

Surgical techniques and perioperative procedures were standardized. Each liver transplantation procedure was performed using a standardized approach of total cava replacement as previously described [28,29]. Standard arterial revascularization was performed as a branch patch anastomosis using the common hepatic artery/splenic artery patch of the donor and the proper hepatic artery/gastroduodenal artery patch of the recipient. If necessary, an aorto-hepatic jump graft revascularization was performed as supra-celiac interposition graft.

The standard immunosuppression regime consisted of basiliximab, tacrolimus, mycophenolate-mofetil, and corticosteroids [28,29]. In cases of suspected MALC in pre-operative imaging with a correlating intraoperative finding (reduced pulse and thrill and/or flow) or in case of a clear MALC related stenosis (more than 50% in diameter with post-stenotic dilatation), the celiac axis was dissected down to the aorta with great attention paid to the surrounding structures, including the pancreas, and the compression was released by surgically dividing the fibrous bands of the MAL on top of the supra-celiac aorta. Restoration of normal flow was confirmed by a strong pulse and thrill throughout the whole respiratory cycle and quantified using a transit time flowmeter (Veri Q, Medistim Germany Ltd., Deisenhofen, Germany). Based on the individual assessment of the transplant surgeon, supported by the results of the intraoperative flowmetry, papaverinhydrochloride was used as topical vasodilatant (Paveron N 25 mg/mL, Linden Arzneimittel-Vertrieb, Heuchelheim, Germany).

Postoperatively, each patient underwent a color Doppler ultrasound every 12 h during the first week or longer. Here the adequate inflow and outflow of the liver were assessed, and Doppler parameters of the hepatic artery hemodynamics were measured. This was combined with a standard blood test including markers of hepatocellular injury (transaminases) and bilirubin. If based on these, the clinical suspicion of a vascular complication (e.g., HAT) was raised, patients were directly submitted to an emergency ceCT angiography to detect potential complications and initiate the timely revision and salvage of the liver graft. If no early complications were observed, patients were monitored daily using a comprehensive standard blood test until discharge from our hospital. If any clinical abnormalities or relevant increase in markers of liver injury were observed anytime during the follow up, further diagnostics, including the exclusion and/or management of vascular complications (color Doppler, ceCT, intervention) were initiated.

If no other indications were present, patients received standard thrombo-embolic prophylaxis for 6 weeks post-OLT and no anti-coagulation therapy was applied.

### 2.4. Contrast-Enhanced CT Detection of Coeliac Axis Stenosis and Other Vascular Alterations

All patients underwent preoperative contrast-enhanced computer tomography (ceCT). Three-dimensional reconstructions were generated with a maximal intensity projection [7]. In our clinical routine all CT scans are examined and demonstrated by a senior staff radiologist during our multidisciplinary liver transplantation board meetings and also by the consultant transplant surgeon immediately before transplantation, to assess vascular abnormalities as well as to plan the reconstruction approach and evaluate the potential need of surgical division of the MAL vs. alternative reconstruction during surgery. Within our study, however, all ceCT examinations were reassessed independently by two experienced staff radiologists (P.I., M.L.) who were blinded for the subsequent surgical treatment approach and postoperative outcomes. Extrinsic stenosis was due to MALC, defined as a characteristic (more than 50% in diameter) superior indentation on the proximal celiac axis usually about 5 mm from its origin at the abdominal aorta with post-stenotic dilatation as described before (Figure 1) [7,9]. Intrinsic stenosis due to atherosclerosis was registered if a significant calcified concentric stenosis was present [7]. Other vascular aberrancies and anatomical variations were registered as well.

### 2.5. European Survey

An online survey instrument using the SurveyMonkey platform (Palo Alto, CA, USA) with open-ended multiple-choice questions was designed by the authors (Z.C., G.L.). Content- and face-validity were initially pilot-tested by five independent senior transplant surgeons. Survey items were updated based on this initial feedback. The survey was formally endorsed by Eurotransplant (Req 553-1.2016). The survey was disseminated online to the clinical leads of 52 European transplant centers between February and April 2017. Although, data reported in this manuscript was part of a more extensive survey research on technical aspects of OLT in Europe [30], the present findings have not been reported before. The original survey questions are available as Appendix A.

### 2.6. Statistical Analysis

Inter-observer agreement for ceCT data was assessed using Cohen`s kappa statistics. Student’s *t* test or Mann–Whitney *U* test were used for comparison of continuous variables, as appropriate. Chi-square test, Fisher’s exact test, and the linear-by-linear association were used for analysis of categorical data, as applicable. The Kaplan–Meier method and the long-rank test were used to plot and compare survival data. Results of the survey were analyzed in a descriptive manner. Statistical analysis was performed using SPSS Statistics version 24 (IBM Corp., Armonk, NY, USA) and data plotted using Prism version 7.0 (GraphPad Software, La Jolla, C, USA).

## 3. Results

Between May 2010 and July 2017, a total of 330 consecutive OLT procedures was performed at the UH-RWTH. After assessing the above-mentioned inclusion- and exclusion criteria, 286 patients were included into the present study.

### 3.1. Imaging Based Preoperative Vascular Evaluation

Blinded vascular evaluation of preoperative ceCTs revealed 149 (52%) patients with no vascular alterations (Table 1). Intrinsic celiac axis stenosis due to atherosclerotic disease was present in 16 (6%) patients (Table 1). Extrinsic stenosis of the celiac axis owing to MALC was found in 34 (12%) patients (Table 1). Portal vein thrombosis was present in 20 (7%) patients (Table 1). Other vascular alterations/aberrancies were present in 93 (33%) cases (Table 1).

Inter-observer agreement for evaluation of vascular abnormalities was very good (median *κ* value 0.90, range 0.78–0.93; Table 1).

### 3.2. Patient Characteristics and Perioperative Outcome

The mean recipient age was 54 ± 10 years, and the majority of patients were male (*n* = 195, 68%). At the time of OLT, the mean laboratory model for end-stage liver disease (MELD)-score was 19 ± 10 and 26 patients (9%) were on life support before OLT. The mean balance of risk (BAR)-score was 8.51 ± 5.45. All recipients received organs from brain dead donors. Some 64% of the donors were classified as ECD [25]. Further characteristics of the study population are presented in Table 2.

Except a slight but statistically significant difference in recipient BMI, no significant disparity was found concerning donor and recipient characteristics between the patients with and without MALC (recipient BMI, *p* = 0.040; Table 2).

Perioperative data, presented in Table 2, did not differ significantly between patients with and without MALC. Twenty-six patients (77%) with MALC underwent standard branch-patch arterial revascularization with successful surgical division of the MAL after recipient hepatectomy (Appendix A). In total, 21 patients (7%) underwent alternative arterial revascularization (e.g., aorto-hepatic jump grafts and revascularizations which were not performed to the branches of the celiac axis in the recipient). No significant difference was found between patients of the MALC and no MALC groups (*p* = 0.274, see Table 2). Twelve patients developed major arterial complications over the first 90-days (5 early HAT, 7 further severe arterial complications). Out of the five HAT cases three resulted in graft loss and re-OLT (60%) and one in graft loss associated with the death of the patient (20%).

Two patients with mild stenosis (≈50% luminal reduction) and initially good intraoperative flow and two cases where MALC was recognized in retrospect during the clinical course did not receive MAL division during OLT (Appendix A). Both patients with the retrospective diagnosis of MALC developed early HAT. One graft was salvaged successfully with open thrombectomy and MALC division. The second patient required re-OLT at POD5, despite a previous revascularization attempt including thrombectomy and MALC division (Appendix A). A third patient with a milder stenosis in imaging and initially good arterial flow intraoperatively and no MAL division developed major graft dysfunction at POD40 with reduced arterial flow to the liver graft which was successfully treated with coil embolization of the splenic artery (Appendix A). These three patients are still alive at the time point of this study with 75-, 70-, and 28-months follow-up respectively. Four MALC patients underwent alternative arterial reconstruction (three aorto-hepatic jump graft; one reconstruction with the aberrant recipient right hepatic artery to the superior mesenteric artery). Two patients in the MALC group died within the first 90-days as results of non-graft related complications (pulmonary embolism; septic shock, and consequential multi organ failure; Appendix A).

### 3.3. Long-Term Follow Up

Five-year patient- and graft survival did not differ significantly between the groups (MALC vs. no MALC; 75% vs. 76%, *p* = 0.837; 72% vs. 73%, *p* = 0.800 respectively) (Figure 2).

### 3.4. European Survey

Forty-two out of the total 52 transplant centers gave a valid response to our survey (81%). Survey responses were received within a mean of 11 ± 12 days. Most of the centers expressed their preference to use ceCT-angiography (28, 67%) in the preoperative assessment of vascular anatomy and abnormalities (Figure 3). This is supported in 15 centers (36%) by color Doppler ultrasound. Only one center (2%) reported the routine use of conventional invasive angiography within the frameworks of pre-operative OLT workup as “other” four centers reported the use of MRI or MRI angiography. Although, half of the respondents (21, 50%) reported that they actively look for MALC and/or other abnormalities of the celiac trunk in the sagittal reconstructions of the CT images, the rest of the centers remarked that they do not actively screen for MALC (10, 24%) or they do so only in cases in which pathology is suspected based on the axial images (10, 24%) (Figure 3).

Twelve centers (29%) had an aggressive approach, routinely dividing the MAL in MALC/MALS cases. According to our survey results, more centers decide to do the surgical division based on the pre-operative assessment using color Doppler ultrasound (3, 7%) or intra-operative flowmetry (15, 36%). Seven respondents (17%) reported that they do not divide the MAL in MALC/MALS, however they carefully follow those patients and decide to correct the occlusion, if necessary (Figure 3). Ten centers (24%) answered that their preferred strategy in case of MALC is to perform alternative e.g., aorto-hepatic reconstructions (Figure 3).

Most of the centers who had a less aggressive approach to the management of MALC felt that the clinical significance of MALC/MALS was controversial (23, 77%), feared injury and complications (6, 20%), and believed to achieve a limited success with potential recurrence (5, 17%) as arguments against the active division of MAL in MALC (Figure 3). One respondent was concerned about causing a pancreatic fistula as a potential complication of MAL dissection.

## 4. Discussion

Median arcuate ligament compression is mostly a diagnosis of exclusion and if symptomatic (MALS) it may cause unspecific abdominal or epigastric symptoms, due to functional ischemia [8,31]. In OLT, MALC may lead to arterial complications requiring an urgent surgical division of the MAL and thrombectomy. In some cases, however, even with immediate actions taken, the liver allograft cannot be saved and re-OLT is inevitable [14,18,19,21].

Until now, only limited evidence of MALC in OLT has been available. Smaller retrospective cohorts report a 1.6%–10% incidence in adult OLT recipients [14,15,17]. In the present study the incidence of MALC was 12% after thorough evaluation of all preoperative ceCT images. A lower incidence of MALC in previous reports may be attributed to the fact that in those studies, diagnosis of MALC was based solely on intraoperative clinical signs of inadequate arterial flow after completion of the arterial anastomosis without signs of calcified stenosis. This approach may be appropriate in identifying MALC patients with clear hemodynamic relevance, but it does not allow the surgeon to plan the surgical approach in advance (e.g., alternative reconstruction vs. standard reconstruction w/ or w/o MAL division). Furthermore, some patients with initially no or little hemodynamic restrictions may be missed and can eventually develop a more severe flow impairment in the postoperative phase with potential factors such as rejection episodes or ischemia-reperfusion injury which might result in acute alterations in parenchymal resistance and hepatic artery blood flow, aggravating the risk of HAT [32].

In our center, we routinely prepare for surgical division of the MAL in all cases when a significant stenosis is suspected in preoperative cross-sectional imaging which is defined as a MAL related stenosis of more than 50% in diameter with post-stenotic dilatation. Only in highly selected cases of mild MAL-related compression but with excellent intraoperative arterial blood-flow, as assessed by intraoperative flow measurements, surgical division of MAL is not performed to avoid an unnecessary risk of surgical injury in this highly sensitive region. In the present study, 34 patients were identified with MALC based on a blinded assessment by two experienced radiologists. No relevant characteristic differences were found concerning pre-operative donor and recipient data, intraoperative factors, including CIT, WIT, type of arterial revascularization, and blood transfusions. In the present cohort, surgical division of MAL was performed in 77% (26/34) and no surgical complications (pancreatic fistula, severe injury of the surrounding structures) related to the division of MAL were observed.

Due to the retrospective nature of this non-randomized analysis and survey study a direct comparison between MAL-division and non-division was not possible. The clinical significance of this condition may, however, be supported by the fact that none of the patients who underwent MAL-division suffered from any early arterial complications, meanwhile two out of five patients who developed HAT over the first 90 days did not undergo MAL-division as MALC was only diagnosed retrospectively (Table 3). Furthermore, one out of the two MALC patients, who did not undergo MAL division because of mild MAL-compression and had an initially good arterial blood flow, was readmitted with allograft dysfunction and liver abscess due to reduced arterial flow and was successfully treated with coil embolization of the splenic artery. In general, no significant differences were found between the MALC and no MALC-groups with regards to early postoperative outcome, including postoperative transfusions of red blood cell units, early allograft dysfunction, morbidity, mortality, ICU- and cumulative hospital stay. Cumulative long-term graft and patient survival were also comparable between the groups.

Although, early major arterial complications including early HAT are rare, they can lead to a rapid deterioration of the allograft function with consecutive graft lost [33,34]. The incidence of early hepatic arterial occlusion after OLT on the basis of HAT ranges between 1.8% to 9% with an over 50% graft loss and re-OLT rate [2,6,35]. In the present study, we found a low rate (2%) of severe HAT in the first 90 days, which however, resulted in a graft loss rate of 80%. As such, the relatively low rate of HAT observed in our study, might also be attributed to the thorough preoperative assessment of vascular aberrancies and careful selection of the surgical approach with aggressive surgical division of the MAL in cases of MALC [15,17,20]. Previous studies have confirmed a wide spectrum of factors that can be associated with early major arterial complications including the type of surgical reconstruction, transfusions, and acute cellular rejection [5,33,34,36,37]. Furthermore, a recent study by Oberkofler et al. suggested that primary arterial patency after liver transplantation is mainly determined by the type of vascular reconstruction rather than patient or disease characteristics [2].

In our OLT cohort, atherosclerosis, as an overall better recognized condition compared to MALC, was responsible for just 32% of relevant celiac axis stenosis. These findings are in line with the previous study of Park et. al., who found that the majority of patients with celiac artery stenosis had MALC and only a few had atherosclerotic stenosis (16 vs. 3) [38]. The degree of celiac artery stenosis varies during respiration and mechanical ventilation and intraoperative hepatic artery flow is influenced by multiple factors, thus determination of a clear clinical flow threshold for division of the MAL is difficult intraoperatively [7]. The present study supports the role of the surgical division of MAL as a safe and effective treatment method if used with the right indication and performed with an appropriate and careful surgical technique [7,19,21].

The clinical significance of MALC is further highlighted by our European survey study (Figure 3). Most of the centers reported the use of high-resolution CT/MR as their first line approach of pre-OLT imaging. In contrast to atherosclerosis, celiac axis stenosis related to MALC is often visible only in sagittal reconstructions [9]. Importantly, only 50% of the surveyed centers reported that they always perform sagittal reconstructions of the CT images and/or actively look for celiac axis aberrancies in the sagittal reconstruction, meanwhile 10 centers responded that they do not actively look for alterations of the celiac axis in sagittal reconstructions (Figure 3). Among these, a few respondents answered that they “do rarely, or do not see MALC patients”, which given the relatively high incidence of this condition in our and other cohorts (12%), suggests a possible under-diagnosis of MALC in OLT patients.

The reported approach on the management of MALC in OLT within the surveyed centers was very heterogeneous. The majority of centers, however, either divide MAL routinely (29%) or are based on intraoperative flowmetry (36%) or perform alternative aorto-hepatic reconstructions in case this condition is present (24%) (Figure 3). The main arguments against the surgical division of MAL were the controversial significance of MALC (77%), followed by the risk of potential injury and complications (20%) as well as the limited success rates (17%). The use of conventional angiography in the routine assessment of potential OLT recipients was only reported by one respondent in our international survey (Figure 3). Based on our experience, angiography does not provide a substantial benefit compared to the ceCT imaging combined with intraoperative assessment, therefore it is not routinely used in our center. There are only very limited data available on preoperative stenting of the celiac axis in potential OLT recipients. As these patients often present with deranged hemostasis due to an impaired liver function, it is important to weigh up the risk of a severe and potentially fatal bleeding (e.g., from oesophageal varices) due to therapeutic anticoagulation following stent placement against the clinical benefits of this intervention. Based on our experience, even if preoperative stenting would be technically possible, in most of the cases the reconstruction following MAL division or using an aorto-hepatic jump graft carry less risk compared to preoperative stenting with good results. Nevertheless, several sporadic reports are available on the post-OLT treatment of hepatic artery thrombosis, due to unrecognized intrinsic stenosis of the celiac axis using endovascular treatment [39,40]. Further studies are warranted to evaluate the exact role of endovascular interventions compare to open revision in the therapeutic algorithms for the stenosis and/or thrombosis of the hepatic artery in OLT recipients.

In adult whole-graft OLT, only a few case reports and a small number of retrospective studies, reporting on over 20 years old patient collectives, are available (Table 3) [15,19,20,21]. Although there is currently no consensus on the treatment of MALC, the main finding of the present study was that the treatment of MALC with MAL division is feasible and safe and, as such, should be considered in all patients with relevant celiac axis stenosis as assessed by preoperative ceCT imaging. These findings are in line with the previous reports, supporting the clinical significance of MALC in OLT as well as the feasibility and high success rate of MAL division in this context [15,19,20,21].

This study has certain limitations. As one of the most important factors, due to the retrospective nature of our assessment, the lack of reliable information on the exact hemodynamic significance of MALC in several cases did not allow a more precise quantified stratification of the severity of stenosis (intraoperative pre- and post-division flow values were not documented in a standard fashion), thus no exact flow values, patterns or flow thresholds could be reported. Likewise, data on other factors such as intraoperative and/or postoperative hypotension, detailed coagulation status were not available. Asymptomatic or milder vascular complications occurring later in the post-transplant course may have remained undetected, and their true incidence might be even higher than reported in this study [2].

Notwithstanding the aforementioned limitations, the present study reports one of the largest patient cohorts assessing the clinical relevance of MALC and the surgical division of MAL in OLT. Even though we could demonstrate a significant heterogeneity and lack of consensus on how centers manage MALC in OLT, the results of our international survey deliver a unique cross-sectional image on how 42 European liver transplantation centers handle these patients. Furthermore, this study also highlights the clinical significance of this poorly investigated condition in OLT. Even though we advocate a rather aggressive approach of surgical MAL-division, prospective validation in larger independent cohorts is warranted.

## Figures and Tables

**Figure 1 jcm-08-00550-f001:**
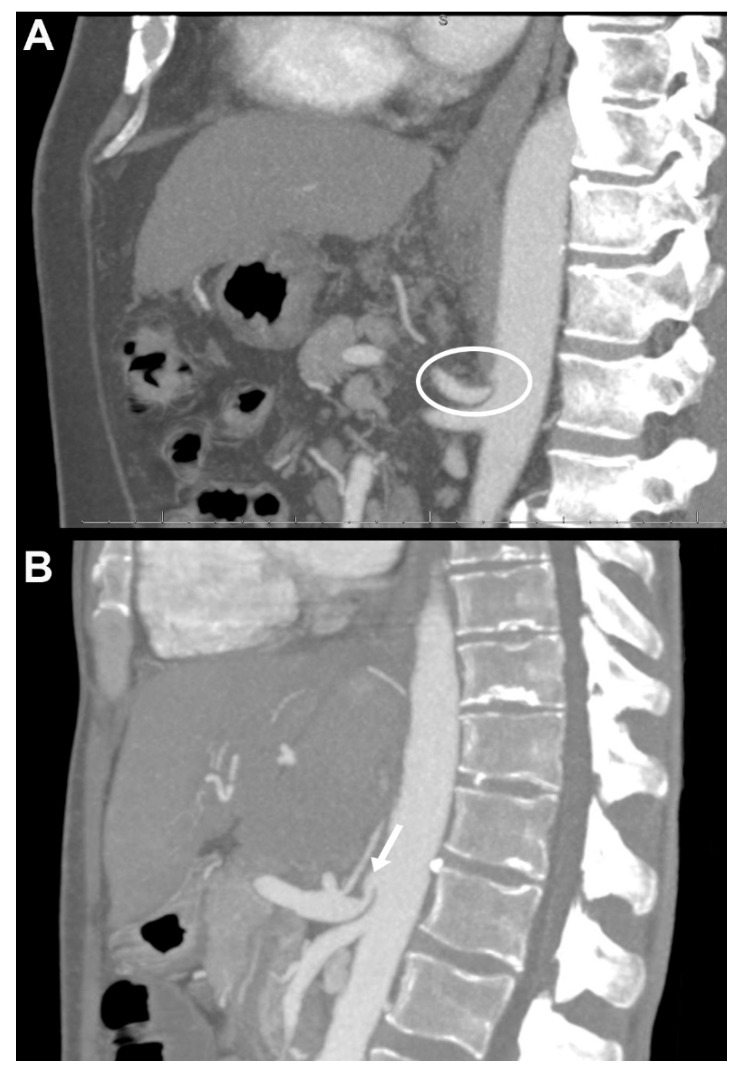
Preoperative computer tomography images of liver transplant recipients with median arcuate ligament compression (MALC) in the sagittal plane using maximal intensity projection. (**A**) Patient with severe median arcuate ligament (MAL) related occlusion. Note the hypertrophic MAL and the lack of continuous contrast in the celiac axis (circle). No calcified plaques were observed. Arterial revascularization was performed using an aorto-hepatic jump graft. (**B**) Patient with MALC. Note the hooked-like appearance of the celiac axis and the post-stenotic dilatation (arrow). MALC was released intraoperatively and standard branch-patch reconstruction could be performed with good outcome and without arterial complications.

**Figure 2 jcm-08-00550-f002:**
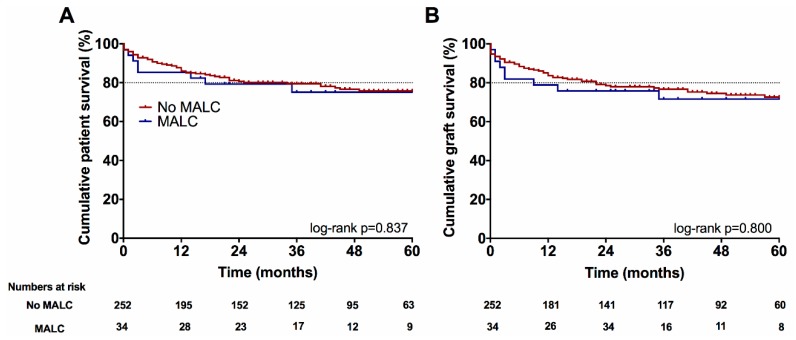
Kaplan–Meier plots for cumulative patient and graft survival Overall patient and graft survival with and without MALC (75% vs. 76%, *p* = 0.837; 72% vs. 73%, *p* = 0.800, respectively). Abbreviations used: MALC, median arcuate ligament syndrome/Dunbar syndrome.

**Figure 3 jcm-08-00550-f003:**
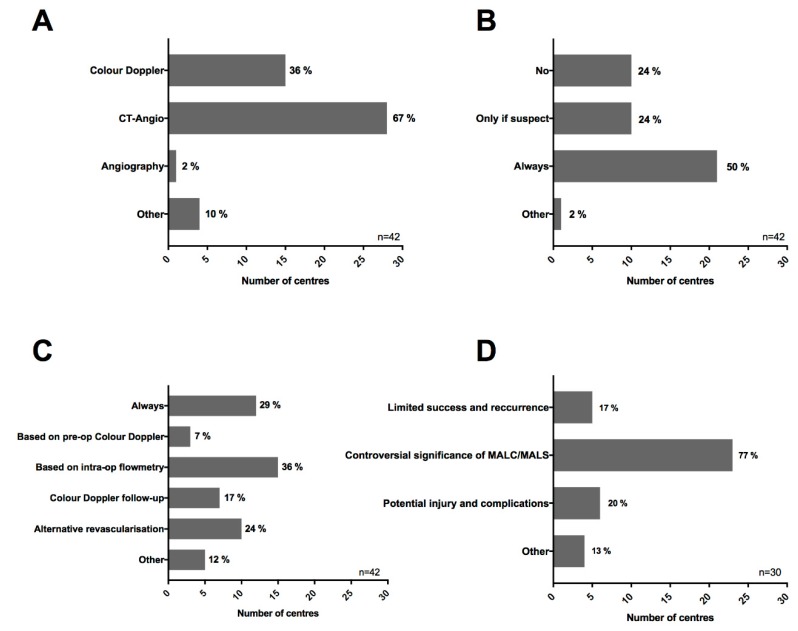
Survey results. Our survey was sent to the clinical leads of 52 European liver transplantation centers. The survey sought information on the local approaches on the diagnostics of vascular aberrancies and the treatment of MALC. Forty-two centers gave a valid response (81%; 42/52). (**A**) Which approach do you routinely use in pre-operative imaging of recipient vascular anatomy? (**B**) Do you routinely look for potential anatomical compression or kinking of the celiac axis in sagittal reconstruction of the CT-scans? (**C**) What is your institutional protocol in cases where you find a suspected anatomical compression of the celiac axis in the pre-operative diagnostics (e.g., Dunbar-syndrome)? (**D**) If you do not routinely divide the MAL in cases of a suspected MAL syndrome what is the main reason behind it? During the assessment of question D, centers which reported at question C that they always divide the MAL in MALC were excluded. Abbreviations used: CT, computed tomography; CD, color Doppler; MALC, median arcuate ligament syndrome/Dunbar syndrome.

**Table 1 jcm-08-00550-t001:** Vascular evaluation based on preoperative contrast-enhanced computer tomographies (ceCTs).

	All Patients (*n* = 286)	*κ*
Normal vascular anatomy	149 (52%)	0.916
Intrinsic celiac axis stenosis/arteriosclerosis	16 (6%)	0.930
Extrinsic celiac axis stenosis/MALC	34 (12%)	0.892
Portal vein thrombosis	20 (7%)	0.848
Other vascular abnormalities	93 (33%)	0.777

Values as given as numbers and (per cent). Kappa represents inter-observer agreement on the radiological diagnosis of various vascular aberrancies in preoperative ceCTs. A *κ* value greater than 0.81 indicated very good agreement, 0.80–0.61 good agreement, 0.60–0.41 moderate agreement, and less than 0.41 indicated poor agreement. Abbreviations used: MALC, median arcuate ligament compression.

**Table 2 jcm-08-00550-t002:** Characteristics of patients who underwent orthotopic liver transplantation (OLT) stratified according to the presence of extrinsic stenosis of the celiac axis (MALC).

	All Patients (*n* = 286)	No MAL Compression (*n* = 252)	MAL Compression (*n* = 34)	*P* Value
Donor age (years)	56 ± 15	56 ± 15	52 ± 17	0.167
Donor BMI	30 ± 7	30 ± 8	28 ± 6	0.146
Donor sex ratio (F:M)	135 (47%):151 (53%)	117 (46%):135 (54%)	18 (53%):16 (47%)	0.475
Pre-transplant Child-Pugh Score	7.4 ± 2.0	7.4 ± 2.0	7.6 ± 2.2	0.683
Donor cause of death	CVA 181 (63%)	CVA 161 (64%)	CVA 20 (59%)	0.565
	Anoxia 60 (21%)	Anoxia 52 (21%)	Anoxia 8 (24%)	0.697
	Trauma 34 (12%)	Trauma 30 (12%)	Trauma 4 (12%)	0.981
	Other 11 (4%)	Other 9 (4%)	Other 2 (6%)	0.626
Extended Criteria Donors ^1^	181 (63%)	162 (64%)	19 (56%)	0.325
Recipient age (years)	54 ± 10	55 ± 11	52 ± 10	0.174
Recipient BMI	27 ± 5	27 ± 5	25 ± 5	0.040
Recipient sex ratio (F:M)	91 (32%):195 (68%)	81 (32%):171 (68%)	10 (29%):24 (71%)	0.748
Etiology of liver disease	ALF 33 (12%)	ALF 26 (10%)	ALF 7 (20%)	0.088
	HCC 70 (24%)	HCC 61 (24%)	HCC 9 (26%)	0.773
	Alcoholic cirrhosis 75 (26%)	Alcoholic cirrhosis 70 (28%)	Alcoholic cirrhosis 5 (15%)	0.104
	Viral 22 (8%)	Viral 18 (7%)	Viral 4 (12%)	0.312
	PSC/PBC 28 (10%)	PSC/PBC 25 (10%)	PSC/PBC 3 (9%)	0.840
	AIH 9 (3%)	AIH 7 (3%)	AIH 2 (6%)	0.291
	Other 49 (17%)	Other 45 (18%)	Other 4 (12%)	0.404
Pre-transplant labMELD	19 ± 10	19 ± 10	20 ± 11	0.697
BAR Score ^2^	8.51 ± 5.45	8.45 ± 5.31	8.76 ± 6.30	0.863
Recipient pre-transplant mechanical/ventilation support ^3^	26 (9%)	22 (9%)	4 (12%)	0.528
Recipient pre-transplant ICU	65 (23%)	56 (22%)	9 (27%)	0.579
Cold ischemic time (min)	503 ± 126	503 ± 123	511 ± 147	0.716
Warm ischemic time (min)	45 ± 8	45 ± 8	45 ± 7	0.996
Intra-operative red blood cell transfusions (units)	9 ± 9	10 ± 10	8 ± 6	0.181
Post-operative red blood cell transfusions (units) ^4^	4 ± 7	4 ± 7	4 ± 6	0.717
Early allograft dysfunction ^5^	80 (28%)	69 (27%)	11 (32%)	0.563
Major complications (≥CD3b) ^6^	145 (51%)	127 (50%)	18 (53%)	0.814
90-days CCI ^7^	52 ± 32	52 ± 32	56 ± 30	0.544
90-days morbidity	CD3 96 (34%)	CD3 82 (33%)	CD3 14 (41%)	0.317
	CD4 78 (27%)	CD4 70 (28%)	CD4 8 (24%)	0.602
	CD5 14 (5%)	CD5 12 (5%)	CD5 2 (6%)	0.676
90-days graft loss	25 (9%)	22 (9%)	3 (9%)	0.542
Alternative arterial revascularization ^8^	21 (7%)	17 (7%)	4 (12%)	0.274
ICU stay (days) ^9^	12 ± 20	12 ± 20	12 ± 16	0.379
Hospital stay (days) ^10^	38 ± 35	39 ± 36	36 ± 23	0.707

Values as given as mean ± standard deviation or numbers and (per cent). ^1^ Refers to German Medical Chamber Guidelines [25], ^2^ Refers to Schlegel et al. [22], ^3^ Mechanical/ventilation support defined as dialysis and/or ventilation before transplantation, ^4^ Refers to blood products given during the first 7-days following OLT, ^5^ Referst to Olthoff et al. [27], ^6^ Refers to Clavien et al. [23], ^7^ Refers to Slankamenac et al. [24], ^8^ Refers to the use of aorto-hepatic jump grafts and revascularizations which were not performed to the celiac axis branches of the recipient, ^9^ Refers to the days spent at the ICU until the first admission to the standard care ward. ICU readmissions are assessed under total hospital stay, ^10^ Hospital stay was defined by the day of discharge from our center to outpatient care and/or rehabilitation. Abbreviations used: BMI, body mass index; F, female; M, male; ECD, extended criteria donors; CVA, cerebrovascular accident; ALF, acute liver failure; HCC, hepatocellular carcinoma; PSC, primary sclerosing cholangitis; PBC, primary biliary cirrhosis, AIH, autoimmune hepatitis; MELD, model for end-stage liver disease; BAR, balance of risk; ICU, intensive care unit; CD, Clavien-Dindo; CCI, comprehensive complications index; ICU, intensive care unit.

**Table 3 jcm-08-00550-t003:** Studies on median arcuate ligament compression (MALC) in orthotopic liver transplantation (search date: 20 March 2019).

First Author	Type of Study	Patients *n* (%) *	Management *n* (%)	Structured Summary
Czigany et al. (present study)	Retrospective cohort	34/286 (12%)	MAL division 26/34 (77), Alternative reconstruction 4/34 (12)	Treatment of MALC with MAL division was feasible and safe and may be considered in patients with relevant celiac axis stenosis as assessed by preoperative ceCT imaging.
Fukuzawa et al., 1993 [20]	Retrospective cohort	5/307 (1.6%)	Aorto-hepatic reconstruction 2/5 (40), MAL division 3/5 (60)	Celiac axis compression resulted in decreased hepatic artery flow to the allograft. In three cases, celiac decompression by division of the arcuate ligament resulted in adequate flow improvement.
Jurim et al., 1993 [17]	Retrospective cohort	17/164 (10%)	MAL division 17/17 (100), Aorto-hepatic reconstruction after unsuccessful MAL division 2/17 (12)	Identification and removal of the obstruction of the celiac axis by MAL division is crucial to prevent severe complications and potential graft loss.
Agnes et al., 2001 [15]	Retrospective cohort	5/140 (3.6%)	MAL division 5/5 (100)	Celiac compression must be identified and corrected in liver transplant recipients to avoid acute or chronic ischemic complications. Surgical therapy consists of arcuate ligament division, usually with excellent prognosis.
Lubrano et al., 2008 [14]	Retrospective cohort	10/168 (6%)	Aorto-hepatic reconstruction 4/10 (40), MAL division 1/10 (10)	The presence of an arcuate ligament does not contraindicate a routine hepatic artery reconstruction and MAL division should be performed if required.

Literature search (PubMed, clinicaltrials.gov; Search terms: liver transplantation AND median arcuate ligament) resulted in 8 relevant reports in adult whole graft OLT focusing on MALC and celiac alterations and MALC. Four case reports were excluded from further evaluation. All four previous retrospective studies report on over 20 year old patient cohorts. However, according to the best of our knowledge our study is one of the largest patient collectives so far on MALC in OLT as well as on the feasibility of surgical division of MAL in MALC during OLT. Despite the scarce evidence, all previous reports agree on the major clinical relevance of MALC in OLT recipients. * MALC patients/All patients in the study (incidence of MALC expressed as per cent). Abbreviations used: MAL, median arcuate ligament; MALC, median arcuate ligament compression.

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
