# Peer review of "Median Arcuate Ligament Compression in Orthotopic Liver Transplantation: Results from a Single-Center Analysis and a European Survey Study"

_jcm, 2019, doi:10.3390/jcm8040550_

Round 1
Reviewer 1 Report
I enjoyed your article as it confirmed my clinical experience in not only liver transplantation but also in pancreatectomy. I was astonished that several surgeons were fearful of cutting the MAL and did not understand the necessity of doing so. My experience has been to look for a MAL if at the time of surgery the pulse in the Common Hepatic Artery was poor. I have recently become cognizant of the use of sagittal CT images to identify MALC but I have not been in the habit of looking for it in potential liver recipients, which I will now do. It would be fascinating if HAT is more often a complication of MALC than not. I do have a slew of editorial comments/suggestions that I feel would clarify many of your points, which I will detail below.
I have several questions about your data:
In Supplementary Table 1, you mention the term "Extrinsic manifest occlusion" in cases #5 and 27. What does this mean? Is this a diagnosis made on the ceCT and means that the celiac axis is occluded? I assume that you had seen on the sagittal images there was clearly a MALC and no blood flow. Thus, I would suggest that you call it "Extrinsic with thrombosis". In those cases have you ever cut the MAL to see if the blood in the celiac axis was not clotted just stagnant?
Also in Supplementary Table 1, you mentioned 2 cases of "Extrinsic, mild stenosis". I was unable to find how you judged "mild stenosis", regular Extrinsic stenosis, "manifest occlusion", and "severe stenosis". As in my question above, why did the surgeon not try to cut the MAL in all cases and only do a bypass if that does not help the inflow. I assume you would now do so in "milder stenosis" based on your experience.
What is the difference between "manifest occlusion" and "severe stenosis"?
In patients #6 and #7, you mentioned that the MALC was identified retrospectively. That begs the question when you actually started doing pre-operative ceCTs? In those patients, did you divide the MAL on reoperation, as that is not mentioned in the Table.
In patient #27, I have tended to use the term "aberrant right hepatic artery" to describe the artery that goes to the right liver and comes off the SMA.
The only difference you found between the patients with MALC and those without is the recipient BMI. I do not feel that is an aberration because MALC is associated with nausea, vomiting, and abdominal pain and thus weight loss.
Line 83: I am confused what you meant by the line "have been assessed within the postoperative complications"? I looked through your paper and did not see any mention of blood products administered after 7 days.
Editorial suggestions:
I by no means consider myself a true Editor or a grammatical expert. I read your paper, trying to determine whether the words you use make sense with what I feel you are trying to say. Clearly, these are suggestions but I feel the changes will improve your paper.
Lines 32-22: Replace "is still" with "remains".
Line 35: Replace "typically featuring" with "caused by".
Line 36: Replace "through" with "by means of".
Line 36: Replace "MAL" with "Median Arcuate Ligament (MAL)". (Possibly, that should be added to the abstract as well.)
Line 37: Replace "mostly" with "usually".
Line 38: Delete "also".
Line 39: Replace "unspecific" with "non-specific".
Line 41: Replace "firstly" with "first".
Line 46: Replace "are" with "were".
Line 48: Replace "have been attributed" with "likely was".
Line 48: Replace "[7,13]" with "[7]".
Line 48: As I did not understand what you meant by the second sentence in this paragraph, I looked at the references and so came up with two sentences that I felt clarify and expand on your points. "Celiac artery stenosis was associated with esophageal conduit necrosis following Ivor-Lewis gastrectomy, of which 50% was due to MALS [7]. MALS was identified on preoperative CT scans in two patients with pancreatic cancer and treated non-invasively in one and surgically in the other; both patients had successful surgeries. [13]."
Line 60: Replace "significant study relevant missing data (lack of sufficient pre-transplant imaging)" with "a lack of sufficient pre-transplant imaging".
Line 61: Rewrite the sentence starting with "Furthermore" to read "Furthermore, patients undergoing simultaneous liver-kidney transplants, those having a living donor liver transplant, and pediatric liver transplants were excluded".
Line 65: Replace "short-long-term" with "short- and long-term".
Line 85: Replace "a" with "the".
Line 156: Replace "in" with "of".
Line 170: Add "mean" after "The" and before "recipient".
Line 178: Delete the period after "2".
Line 183: Replace "grafts" with "graft".
Line 185: Replace "were" with "was".
Line 186: Delete "have".
Line 187: Delete "an".
Line 189: Delete "has".
Line 190: Delete "a".
Line 227: Replace "have been" with "were".
Line 227: Insert "of" after "mean".
Line 235: Replace "if there is a pathology" with "in which pathology is".
Line 236: Replace "stated" with "had".
Line 236: Replace "reported the routine divisions of" with "routinely divided the".
Line 237: Add "of the" after "More".
Line 237: Replace "decide for the" with "decided to do".
Line 237: Add "the" after "on".
Line 238: Replace "with" with "using".
Line 239: Add "carefully" after "they" and before "follow".
Line 240: Replace "these patients strictly and decide for revision if necessary" with "those patients and decide to correct the occlusion, if necessary.".
Line 243: I suggest rewriting that first sentence to "Most of the centers who had a less aggressive approach to the management of MALC felt that the clinical significance of MALC/MALS was controversial (23, 77%), feared injury and complications (6, 20%), and felt there was limited success and potential recurrence (5, 17%) as arguments against the active division of MAL in MALC (Figure 3).".
Line 246 and 247: Replace "has specified the fear of" with "was concerned about causing a".
Line 274: Replace "with the need of" with "requiring".
Line 277: Replace "on" with "of".
Line 286: Delete "out".
Line 295: Replace "preparation" with "injury".
Line 296: Replace "have" with "had".
Line 307: Replace "in a retrospective work-up of these cases" with "retrospectively".
Line 308: Insert "had" between "and" and "initially".
Line 319: Replace "converted into an 80% graft loss rate" with "resulted in a graft loss rate of 80%".
Line 324: Add "and" after "transfusions,".
Line 328: I suggest rewriting the sentence beginning with "These findings" wo "These findings are in line with the previous study of Park, et. al., who found that the majority of patients with celiac artery stenosis had MALC and only a few had atherosclerosis (16 patients vs 3) [40]."
Line 331: Rewrite the sentence beginning with "Celiac axis" to "The degree of celiac artery stenosis varies during respiration and mechanical ventilation and intraoperative hepatic artery flow is influenced by multiple factors, thus determination of a clear clinical flow threshold for division of the MAL is difficult intraoperatively [7]."
Line 333: Delete "a".
Line 342: Add "a" before "few respondents".
Line 343: Delete "as free text response".
Line 362: Add "and a".
Line 364: Replace "as" with "than".
Line 365: Replace "collective' with "study".
Line 366: Replace "assessing" with "of".
Line 366: Add "of" and "and" and before "surgical".
In Supplementary Table 1, please have the terms look the same, i.e., you have listed patient #5 as having "Extrinsic manifest occlusion" and patient #27 as having "Extrinsic; manifest occlusion". I would suggest in the 7 cases where you have modified either Extrinsic or Intrinsic, that the modifying phrase be written in the form "Extrinsic, severe stenosis".
In that same table, Pseudoaneurism should be spelled pseudoaneurysm.
Author Response
Point-to-point answers to Reviewer 1 comments:
I enjoyed your article as it confirmed my clinical experience in not only liver transplantation but also in pancreatectomy. I was astonished that several surgeons were fearful of cutting the MAL and did not understand the necessity of doing so. My experience has been to look for a MAL if at the time of surgery the pulse in the Common Hepatic Artery was poor. I have recently become cognizant of the use of sagittal CT images to identify MALC but I have not been in the habit of looking for it in potential liver recipients, which I will now do. It would be fascinating if HAT is more often a complication of MALC than not. I do have a slew of editorial comments/suggestions that I feel would clarify many of your points, which I will detail below.
We appreciate the expert comments underlining the clinical importance of MALC, not only in orthotopic liver transplantation (OLT) but also in a wide spectrum of major surgical interventions of the upper abdomen.
I have several questions about your data: In Supplementary Table 1, you mention the term "Extrinsic manifest occlusion" in cases #5 and 27. What does this mean? Is this a diagnosis made on the ceCT and means that the celiac axis is occluded? I assume that you had seen on the sagittal images there was clearly a MALC and no blood flow. Thus, I would suggest that you call it "Extrinsic with thrombosis". In those cases have you ever cut the MAL to see if the blood in the celiac axis was not clotted just stagnant?
Also in Supplementary Table 1, you mentioned 2 cases of "Extrinsic, mild stenosis". I was unable to find how you judged "mild stenosis", regular Extrinsic stenosis, "manifest occlusion", and "severe stenosis". As in my question above, why did the surgeon not try to cut the MAL in all cases and only do a bypass if that does not help the inflow. I assume you would now do so in "milder stenosis" based on your experience.
What is the difference between "manifest occlusion" and "severe stenosis"?
We thank the reviewer for drawing attention to this important issue. We fully agree and revised our supplementary files accordingly. Extrinsic manifest occlusion was indeed characterized by the sign of a complete lack of continuous contrast in the celiac trunk with severe MALC related stenosis (this is normally seen on a very short segment due to retrograde perfusion and contrast of the post stenotic part. We have revised these cases in our supplementary table and called them “Extrinsic; manifest occlusion with thrombosis”. In our clinical routine, in these cases we also clinically evaluate the blood flow of the celiac trunk, however, in our experience severe manifest occlusion due to MALC normally combined with a hypoplastic vessel wall in the stenotic area which normally does not recover well after dividing the MALC. Therefore, in these severe cases an aorto-hepatic reconstruction using a jump graft is the treatment of choice in our practice.
We would like to apologize for the partially confusing classification of the extent of extrinsic stenosis in MALC patients within our supplementary table 1. As already mentioned above, severe cases of MALC with occlusion and thrombosis in imaging has been revised as “Extrinsic; manifest occlusion with thrombosis” in our table. One case (case 32) has been classified as extrinsic severe stenosis. In this patient we could see an expressed stenosis, however without clear indication of manifest thrombosis of the lumen. Intraoperatively this patient had a hypoplastic celiac trunk, therefore the decision was made to perform a primary aorto-hepatic reconstruction without attempting conventional reconstruction and MAL division. To further indicate this fact, we have provided this information in the supplementary table 1 for patient 32. A more interesting question is (as indicated by the referee) the treatment of choice for patients with clear but milder stenosis. The term “milder stenosis” was used in two cases (cases 26, 29). As mentioned in our results (Line 207-217) these patients had a stenosis which was at our pre-defined diagnostic threshold (@50% luminal reduction) in imaging. Both of these patients have presented with initially good arterial pulse intraoperatively, therefore the surgeon has omitted MAL division in these cases. Although, patient 29 had an uneventful recovery, patient 26 has developed a major graft dysfunction during the relatively early post-OLT phase (POD40) with reduced arterial flow to the liver graft which was successfully treated with coil embolization of the splenic artery. These observations are probably in line with the suggestion of the reviewer, to be even more radical in MALC patients and divide the MAL in all cases when a relevant stenosis can be observed in imaging even if there is a seemingly good flow during hepatectomy. Some patients with milder stenosis of around 50% or even higher luminal reduction and initially no or little hemodynamic restrictions may develop a more severe flow impairment in the postoperative phase due to potential factors such as rejection episodes or ischemia-reperfusion injury which might result in acute alterations in parenchymal resistance and hepatic artery blood flow, aggravating the risk of complications.
In patients #6 and #7, you mentioned that the MALC was identified retrospectively. That begs the question when you actually started doing pre-operative ceCTs? In those patients, did you divide the MAL on reoperation, as that is not mentioned in the Table.
We routinely perform preoperative imaging (CTA or alternatively in very few cases MRA) in potential OLT recipients. Patients which were excluded due to the lack of sufficient imaging were mostly external referrals for high urgency listing and/or patients with external imaging of ambivalent quality (e.g. only venous phase) which has not been repeated in domo. However, all patients who were included in the study undergone preoperative ceCT imaging (see “Patients and methods”). As mentioned in our methods section, in the clinical routine all CT scans are examined and demonstrated by a senior staff radiologist during our multidisciplinary liver transplantation board meetings, to assess vascular abnormalities as well as to plan the reconstruction approach in advance and evaluate the potential need of surgical division of the MAL vs. alternative reconstruction during surgery. Within our study, however, all ceCT examinations were reassessed independently by two experienced staff radiologists who were blinded for the subsequent surgical treatment approach and postoperative outcomes. After this we could identify the two cases mentioned by the reviewer, MALC was unfortunately NOT documented in the preoperative evaluation. The condition in these patients was only recognized retrospectively, after complications have occurred in these recipients. As it is a retrospective study, we were not able to recover the reason which led to the fact that MALC was not documented in these recipients and no MAL division was performed. In both cases MALC was divided at the time of the surgical revision and thrombectomy. Our supplementary table 1 has been revised to include this information as suggested by the reviewer.
In patient #27, I have tended to use the term "aberrant right hepatic artery" to describe the artery that goes to the right liver and comes off the SMA.
We would like to thank the reviewer for this valuable expert comment. The supplementary table 1 has been revised accordingly.
The only difference you found between the patients with MALC and those without is the recipient BMI. I do not feel that is an aberration because MALC is associated with nausea, vomiting, and abdominal pain and thus weight loss.
As the reviewer suggested, we do not consider the significant difference in recipient BMI as a major factor in this context. Nonetheless, it has been described that MALC is more frequently observed in women (4:1 ratio, which could not be reproduced in the present study) and more patients are having a lean body type [1]. At this point it is very important, however, to clearly differentiate between the definitions of median arcuate ligament compression and median arcuate ligament syndrome (MALS). Patients with manifest MALS may present with nausea, vomiting, and non-specific abdominal pain and thus weight loss. Meanwhile, MALC is “only” an anatomical condition which is often subclinical and only relevant following major abdominal surgeries if the rich abdominal collaterals are divided and the stenotic celiac trunk will be the bottleneck of tissue perfusion in the postoperative phase. In most of the surgical candidates preoperative MALC is present without manifest MALS. In our study non of our patients with MALC had treated or identified with the diagnosis of clinically manifest MALS in the preoperative phase according to our clinical records, however it is unclear whether any of them had non-specific abdominal symptoms associated with MALC which remained unrecognized by the referring hepatologists or have been assessed as consequence of the end stage liver disease. Therefore, in most of the cases the only way to recognize this subclinical anatomical condition is to actively look for it preoperatively.
Line 83: I am confused what you meant by the line "have been assessed within the postoperative complications"? I looked through your paper and did not see any mention of blood products administered after 7 days.
We would like to apologize for the confusing description. Patients who have received transfusion of blood products later on during the first 90-days postoperatively but after the 7. POD, the blood transfusion itself was classified as Clavien-Dindo Grade II complication, as described and recommended by Clavien et al. [2]. These blood products have not been mentioned numerically in table 2 as numbers of units administered, but have only been registered within the Clavien-Dindo classification and the CCI scores to provide a comprehensive assessment of all complications and relevant interventions over the first 90-days following OLT [2, 3]. To clarify this issue the corresponding sentence of the methods section has been revised accordingly (Line 85-87).
Editorial suggestions: I by no means consider myself a true Editor or a grammatical expert. I read your paper, trying to determine whether the words you use make sense with what I feel you are trying to say. Clearly, these are suggestions but I feel the changes will improve your paper.
Here, we would like to express our gratitude for the thorough and thoughtful review of the referee and for the suggestions which have clearly improved the readability and quality of our manuscript.
Lines 32-22: Replace "is still" with "remains".
Line 35: Replace "typically featuring" with "caused by".
Line 36: Replace "through" with "by means of".
Line 36: Replace "MAL" with "Median Arcuate Ligament (MAL)". (Possibly, that should be added to the abstract as well.)
Line 37: Replace "mostly" with "usually".
Line 38: Delete "also".
Line 39: Replace "unspecific" with "non-specific".
Line 41: Replace "firstly" with "first".
Line 46: Replace "are" with "were".
Line 48: Replace "have been attributed" with "likely was".
Line 48: Replace "[7,13]" with "[7]".
Line 48: As I did not understand what you meant by the second sentence in this paragraph, I looked at the references and so came up with two sentences that I felt clarify and expand on your points. "Celiac artery stenosis was associated with esophageal conduit necrosis following Ivor-Lewis gastrectomy, of which 50% was due to MALS [7]. MALS was identified on preoperative CT scans in two patients with pancreatic cancer and treated non-invasively in one and surgically in the other; both patients had successful surgeries. [13]."
The above points have been revised and changed according to the recommendations of the reviewer.
Line 60: Replace "significant study relevant missing data (lack of sufficient pre-transplant imaging)" with "a lack of sufficient pre-transplant imaging".
Line 61: Rewrite the sentence starting with "Furthermore" to read "Furthermore, patients undergoing simultaneous liver-kidney transplants, those having a living donor liver transplant, and pediatric liver transplants were excluded".
Line 65: Replace "short-long-term" with "short- and long-term".
Line 85: Replace "a" with "the".
Line 156: Replace "in" with "of".
Line 170: Add "mean" after "The" and before "recipient".
Line 178: Delete the period after "2".
Line 183: Replace "grafts" with "graft".
Line 185: Replace "were" with "was".
Line 186: Delete "have".
Line 187: Delete "an".
Line 189: Delete "has".
Line 190: Delete "a".
Line 227: Replace "have been" with "were".
Line 227: Insert "of" after "mean".
Line 235: Replace "if there is a pathology" with "in which pathology is".
Line 236: Replace "stated" with "had".
Line 236: Replace "reported the routine divisions of" with "routinely divided the".
Line 237: Add "of the" after "More".
Line 237: Replace "decide for the" with "decided to do".
Line 237: Add "the" after "on".
Line 238: Replace "with" with "using".
Line 239: Add "carefully" after "they" and before "follow".
Line 240: Replace "these patients strictly and decide for revision if necessary" with "those patients and decide to correct the occlusion, if necessary.".
Line 243: I suggest rewriting that first sentence to "Most of the centers who had a less aggressive approach to the management of MALC felt that the clinical significance of MALC/MALS was controversial (23, 77%), feared injury and complications (6, 20%), and felt there was limited success and potential recurrence (5, 17%) as arguments against the active division of MAL in MALC (Figure 3).".
Line 246 and 247: Replace "has specified the fear of" with "was concerned about causing a".
Line 274: Replace "with the need of" with "requiring".
Line 277: Replace "on" with "of".
Line 286: Delete "out".
Line 295: Replace "preparation" with "injury".
Line 296: Replace "have" with "had".
Line 307: Replace "in a retrospective work-up of these cases" with "retrospectively".
Line 308: Insert "had" between "and" and "initially".
Line 319: Replace "converted into an 80% graft loss rate" with "resulted in a graft loss rate of 80%".
Line 324: Add "and" after "transfusions,".
Line 328: I suggest rewriting the sentence beginning with "These findings" wo "These findings are in line with the previous study of Park, et. al., who found that the majority of patients with celiac artery stenosis had MALC and only a few had atherosclerosis (16 patients vs 3) [40]."
Line 331: Rewrite the sentence beginning with "Celiac axis" to "The degree of celiac artery stenosis varies during respiration and mechanical ventilation and intraoperative hepatic artery flow is influenced by multiple factors, thus determination of a clear clinical flow threshold for division of the MAL is difficult intraoperatively [7]."
Line 333: Delete "a".
Line 342: Add "a" before "few respondents".
Line 343: Delete "as free text response".
Line 362: Add "and a".
Line 364: Replace "as" with "than".
Line 365: Replace "collective' with "study".
Line 366: Replace "assessing" with "of".
Line 366: Add "of" and "and" and before "surgical".
In Supplementary Table 1, please have the terms look the same, i.e., you have listed patient #5 as having "Extrinsic manifest occlusion" and patient #27 as having "Extrinsic; manifest occlusion". I would suggest in the 7 cases where you have modified either Extrinsic or Intrinsic, that the modifying phrase be written in the form "Extrinsic, severe stenosis".
In that same table, Pseudoaneurism should be spelled pseudoaneurysm.
We would like to kindly thank the referee for these suggestions which we have attempted to fully implement in the revised version of our paper.
References:
1. Kim EN, Lamb K, Relles D, Moudgill N, DiMuzio PJ, Eisenberg JA. Median Arcuate Ligament Syndrome-Review of This Rare Disease. JAMA surgery. 2016;151(5):471-7.
2. Clavien PA, Barkun J, de Oliveira ML, Vauthey JN, Dindo D, Schulick RD, de Santibanes E, Pekolj J, Slankamenac K, Bassi C, Graf R, Vonlanthen R, Padbury R, Cameron JL, Makuuchi M. The Clavien-Dindo classification of surgical complications: five-year experience. Annals of surgery. 2009;250(2):187-96.
3. Slankamenac K, Graf R, Barkun J, Puhan MA, Clavien PA. The comprehensive complication index: a novel continuous scale to measure surgical morbidity. Annals of surgery. 2013;258(1):1-7.
Reviewer 2 Report
The authors are to be commended for their thorough evaluation and treatment of MALC, which as they describe is often a diagnosis of exclusion. Clinical significance of MALC is potentially high if related to increased arterial complications, which have devastating effects in liver transplant recipients.
After reading the manuscript, I have few remaining questions:
- Method section: the authors describe two indications for MAL division:
1. In cases of suspected MALC in pre-operative imaging with a correlating intraoperative finding (flow measurement) or 2. in case of a clear MALC related stenosis (more than 50% in diameter with post-stenotic dilatation), the celiac axis was dissected down to the aorta
It is unclear from the result section how many patients were in group 1 and in group 2. It is unclear if a third group of patients could be identified: cases of suspected MALC in pre-operative imaging with no correlating intra-operative finding.
- It may be helpful to present the results of primary outcomes in MALC patients compared to non-MALC recipients: how many alternative arterial reconstructions, how many arterial complications, in both groups. Is there a significant difference?
- As no comparison can be made between yes/no MALC division, it is unclear whether MALC division does contribute to the outcomes.
- No outcomes of intra-operative flow measurement in the total patient group was described
- No standard postoperative follow-up for detection of arterial complications was described (was standard CT or ultrasound performed)? No definition of major arterial complications was given.
- What was the standard protocol for postoperative anti-coagulation therapy?
- Table 3 could be improved by structuring the table into numbers, not texts. E.g. columns MAL n/%. MAL division n/%. Alternative reconstructions n/%. HAT MAL vs. non-MAL recipients. Conclusion: structured summaries, no citations. Please, leave out case reports, they deserve no place in a table. NB 5/140 = 3.6% NOT 36%
- Table 2 may be shortened to contain most relevant parameters or could be moved to supplementary materials
- The addition of a separate group (intrinsic celiac axis stenosis) does in my opinion not increase significance or readability of the manuscript
- Alternative ways to deal with MALC recipients and identification of significant stenosis (preoperative angiography with flow measurement, stent placement) have not been discussed. Was this approach considered or done in few cases?
Author Response
Point-to-point answers to Reviewer 2 comments:
The authors are to be commended for their thorough evaluation and treatment of MALC, which as they describe is often a diagnosis of exclusion. Clinical significance of MALC is potentially high if related to increased arterial complications, which have devastating effects in liver transplant recipients. After reading the manuscript, I have few remaining questions:
Method section: the authors describe two indications for MAL division:
1. In cases of suspected MALC in pre-operative imaging with a correlating intraoperative finding (flow measurement) or 2. in case of a clear MALC related stenosis (more than 50% in diameter with post-stenotic dilatation), the celiac axis was dissected down to the aorta. It is unclear from the result section how many patients were in group 1 and in group 2. It is unclear if a third group of patients could be identified: cases of suspected MALC in pre-operative imaging with no correlating intra-operative finding.
The authors would like to express their gratitude for the thorough reviews and expert suggestions of the referee which we have implemented in the revised version of our manuscript. Within our study, extrinsic stenosis due to MALC was defined as a characteristic (more than 50% in diameter) superior indentation on the proximal celiac axis usually about 5mm from its origin at the abdominal aorta with post-stenotic dilatation (see Figure 1) as described by other authors before [1]. As shown in our supplementary table 1, in all cases, except patient 6 and 7, where the diagnosis of MALC was made retrospectively, the finding of the imaging was correlated with the clinical evaluation of the transplanting surgeon. If the occlusion was manifest with or without thrombosis in imaging and/or with hypoplastic celiac trunk, alternative aorto-hepatic reconstruction was performed using a jump graft. In all other cases MALC division was performed and restoration of normal flow was confirmed by a strong pulse and thrill throughout the whole respiratory cycle supported by transit time flowmetry.
Only two patients were identified with MAL related stenosis at the diagnostic threshold (@50% luminal reduction) in imaging (see also Reviewer 1 comments, Question and Answer 2). These cases would belong to a suggested “third group” as mentioned by the reviewer where pre-operative imaging showed no clear correlation with the intra-operative finding. Both of these patients have presented with initially good arterial pulse intraoperatively, therefore the surgeon has omitted MAL division in these cases. Although, patient 29 had an uneventful recovery, patient 26 has developed a major flow related graft dysfunction later. Despite these slight variation in terms of the severity of MALC we have attempted to use clearly defined imaging criteria to identify patients with MALC. Based on the limited case number, due to the relative rarity of this condition, we do not think that further subdivision of this group of patients would have improved the quality of the results and would have strengthened the message of the paper. However, to further improve the clarity of our definitions in terms of diagnostic criteria and patient identification, our methods section (Line 103-108) as well as the supplementary table 1 have been revised.
It may be helpful to present the results of primary outcomes in MALC patients compared to non-MALC recipients: how many alternative arterial reconstructions, how many arterial complications, in both groups. Is there a significant difference?
This is indeed an important point which we would like to clarify further in our manuscript. To provide more information on the type of arterial reconstruction, as suggested by the referee, the numbers of patients with alternative arterial revascularization (e.g. aorto-hepatic jump grafts and reconstructions which were not performed to the celiac axis branches of the recipient) have been included into table 2. Nevertheless, due to the low incidence of early HAT (less than 2% and no observed case in the patients underwent MALC division and standard revascularization) and the lack of a no MAL division group of MALC patients, no statistically meaningful comparison was possible in the context of hepatic artery thrombosis.
As no comparison can be made between yes/no MALC division, it is unclear whether MALC division does contribute to the outcomes.
We appreciate this comment of the reviewer drawing attention to this important topic. As it is mostly the case for studies with similar “surgical” research questions, it is difficult to design a comparative prospective study which would allow a clear statement, whether MAL division is superior compared to non-division. According to the best of our knowledge no prospective study is available which would make this direct comparison in OLT or in any other surgical intervention where MALC might play a crucial role in the outcome (e.g. surgery of the pancreas or upper GI surgery). This can be attributed to various factors which make that study design unfeasible. One is certainly the relative rarity of the disease and the lack of clear diagnostic consensus. Because of this, for this specific research question (MAL division vs. no division) one would require a multi-center design to achieve a satisfactory statistical power for the incidence of hepatic artery thrombosis as primary endpoint. This would require the complete standardization of the surgical technique and every other potential cofounding factors among the participating centers (e.g. immunosuppression or transfusion policies) which are currently highly heterogeneous between surgical teams and mostly based on local protocols and personal experience of the team [2]. It should be mentioned, however, that previous studies have also successfully implemented a similar retrospective study design to explore the relevance of celiac trunk stenosis in other conditions such as in patients undergoing curative resection for oesophageal cancer (e.g. Lainas, British Journal of Surgery, 2017) [3].
Therefore, our aim was not to directly show whether division of MALC in OLT is superior compared to the non-division. The main finding of our study was that the treatment of MALC with MAL division is feasible and safe and, as such, should be considered in these highly selected patients with relevant extrinsic celiac axis stenosis as assessed by preoperative ceCT imaging. These findings are basically also in line with previous reports, supporting the clinical significance of MALC in OLT as well as the feasibility and good success rate of MAL division in this condition (Table 3). To further improve clarity on the research aims of our assessment, we have slightly revised the introduction of our paper.
Currently available evidence on MALC in OLT is mostly limited to a handful of reports which are decades old and to few case reports (Table 3). However, multiple quality reports have emerged over the last couple of years, indicating the clinical importance of MALC in various other surgical interventions of the upper abdomen such as oesophagectomies and pancreato-duodenectomies [3, 4], therefore we believe that, despite certain obvious limitations, our present report on MALC in OLT including a retrospective clinical assessment as well as an international survey study is important in raising awareness and further interest for this unique condition in these complex group of patients.
No outcomes of intra-operative flow measurement in the total patient group was described
As we have declared in our discussion (Line 394-402) this as one of the most important limitation of our present work, due to the retrospective nature of our assessment, we were not able to reliably retrieve all information on the exact hemodynamic significance of MALC which did not allow a more precise quantified stratification of the severity of stenosis. However, measurement of the hepatic artery flow using a transit time flowmeter (Veri Q, Medistim Germany Ltd., Deisenhofen, Germany) belongs to the clinical routine in our department, intraoperative pre- and post-division flow values were not documented in a standardized fashion for the study period, therefore they were omitted from the present report. For future studies we have newly established a way of standardized documentation of the flow measurements. To further emphasize this factor as limitation in our paper, we have revised the corresponding sentence of our discussion.
No standard postoperative follow-up for detection of arterial complications was described (was standard CT or ultrasound performed)? No definition of major arterial complications was given.
The reviewer raises an issue of great clinical relevance which was indeed not carefully outlined in the original version of our manuscript. In our clinical routine all patients receive color Doppler ultrasound every 12 hours for the first 7 days (or depending on the clinical situation eventually longer), performed by an experienced transplant fellow. Here the adequate inflow and outflow of the liver are assessed, and Doppler parameters of the hepatic artery hemodynamics are measured (velocity and resistance index). This is combined with a standard blood test including markers of hepatocellular injury (transaminases) and bilirubin. If based on these, the clinical suspicion of a vascular complications including HAT is raised, patients are directly submitted to an emergency CT angiography to detect potential HAT and initiate the timely revision of the patient to salvage the liver graft. If no complications are observed in this initial phase, patients receive a comprehensive standard blood test daily until discharge from our hospital. If any clinical abnormalities or relevant increase in markers of liver injury are observed, further diagnostics including the exclusion of vascular complications (color Doppler, CTA) are initiated. Following discharge from the hospital, patients regularly visit our outpatient clinic and laboratory tests are performed and sent to our transplant hepatologist by the GP 2-3 times a week. To provide more information on these practices and better define major arterial complications we have revised our manuscript accordingly (Line 114-126 and Line 81-84).
What was the standard protocol for postoperative anti-coagulation therapy?
In standard cases of MALC division and conventional arterial reconstruction no further intervention was performed. The question is more complex in cases with alternative revascularization (e.g. jump graft). Although, aorto-hepatic reconstruction is seemingly associated with an increased risk of HAT compared to conventional revascularization, there is no clear evidence or consensus concerning the beneficial effects of a routine anti-coagulation therapy or aspirin prophylaxis in these patients [5], therefore, if no other indication was present, our patients have received standard thrombo-embolic prophylaxis for 6 weeks post-OLT but no anti-coagulation therapy was applied. To include this information, we have revised our manuscript accordingly (Line 125-126).
Table 3 could be improved by structuring the table into numbers, not texts. E.g. columns MAL n/%. MAL division n/%. Alternative reconstructions n/%. HAT MAL vs. non-MAL recipients. Conclusion: structured summaries, no citations. Please, leave out case reports, they deserve no place in a table. NB 5/140 = 3.6% NOT 36%
We would like to thank the reviewer for his/her constructive suggestion. Table 3 has been revised to improve clarity as suggested and case reports have been removed from the table. However, in most of the previous papers no data on the exact numbers of alternative revascularizations were available for the non-MAL patients, therefore this could not be reported in our table. We would also like to apologize for the numerical mistake (3.6 vs. 36%) made and identified by the referee.
Table 2 may be shortened to contain most relevant parameters or could be moved to supplementary materials
Table 2 has been revised and shortened accordingly, however, still retaining the clinically more relevant parameters.
The addition of a separate group (intrinsic celiac axis stenosis) does in my opinion not increase significance or readability of the manuscript
The detailed description of patients with intrinsic stenosis have been removed from our supplementary table 1 and results, however to retain the comprehensive nature of our manuscript, the pure numerical data related to these patients have been kept in our table 1.
Alternative ways to deal with MALC recipients and identification of significant stenosis (preoperative angiography with flow measurement, stent placement) have not been discussed. Was this approach considered or done in few cases?
We cordially thank the referee for this important aspect, with which we fully concur. Although, conventional (invasive) angiography is considered as obsolete in the standard evaluation of OLT recipients, it might have a special role to play in selected patients where intrinsic or extrinsic stenosis is suspected based on ceCT imaging. In our international survey only one respondent has marked that angiography still represents the clinical standard in the diagnostic evaluation of these patients in their center. Based on our experience, angiography is not required in the standard cases of MALC patients, it might deliver important information on the preoperative hemodynamic situation in some selected complex cases. As OLT recipients are often present with hemostatic alterations due to an impaired liver function, it is important to weigh up the risk of a severe and potentially fatal bleeding (e.g. from esophageal varices) due to therapeutic anticoagulation following stent placement and the benefit obtained by this intervention. Based on our experience, even if preoperative stenting would be technically possible, in most of the cases the reconstruction following MAL division or using an aorto-hepatic jump graft carries less risk and is technically more feasible compared to preoperative stenting. However, multiple reports are available on the post-OLT treatment of hepatic artery thrombosis, due to unrecognized stenosis of the celiac axis using endovascular treatment [6, 7]. Endovascular stenting is, however, indicated mostly if the etiology of the stenosis is intrinsic obstruction. Severe extrinsic compression can lead to stent crushing and poor clinical outcome with stent re-occlusion [7]. As time is a crucial factor in post-OLT arterial complications for liver graft salvage, especially during the early post-OLT days and weeks when HAT is associated with very high probability of graft loss, we opt for open revision and thrombectomy combined with MAL division (if not performed during the initial surgery) as it provides a rapid and radical solution for this life-threatening and dramatic complication. The discussion of our manuscript has been revised, to include the value and limitations of these potential alternative ways to deal with MALC in OLT as suggested by the referee (Line 371-386).
References:
1. Horton KM, Talamini MA, Fishman EK. Median arcuate ligament syndrome: evaluation with CT angiography. Radiographics : a review publication of the Radiological Society of North America, Inc. 2005;25(5):1177-82.
2. Czigany Z, Scherer MN, Pratschke J, Guba M, Nadalin S, Mehrabi A, Berlakovich G, Rogiers X, Pirenne J, Lerut J, Mathe Z, Dutkowski P, Ericzon BG, Malago M, Heaton N, Schoning W, Bednarsch J, Neumann UP, Lurje G. Technical Aspects of Orthotopic Liver Transplantation-a Survey-Based Study Within the Eurotransplant, Swisstransplant, Scandiatransplant, and British Transplantation Society Networks. Journal of gastrointestinal surgery : official journal of the Society for Surgery of the Alimentary Tract. 2018.
3. Lainas P, Fuks D, Gaujoux S, Machroub Z, Fregeville A, Perniceni T, Mal F, Dousset B, Gayet B. Preoperative imaging and prediction of oesophageal conduit necrosis after oesophagectomy for cancer. The British journal of surgery. 2017;104(10):1346-54.
4. Sugae T, Fujii T, Kodera Y, Kanzaki A, Yamamura K, Yamada S, Sugimoto H, Nomoto S, Takeda S, Nakao A. Classification of the celiac axis stenosis owing to median arcuate ligament compression, based on severity of the stenosis with subsequent proposals for management during pancreatoduodenectomy. Surgery. 2012;151(4):543-9.
5. Oberkofler CE, Reese T, Raptis DA, Kummerli C, de Rougemont O, De Oliveira ML, Schlegel A, Dutkowski P, Clavien PA, Petrowsky H. Hepatic artery occlusion in liver transplantation - What counts more: Type of reconstruction or severity of recipient's disease? Liver transplantation : official publication of the American Association for the Study of Liver Diseases and the International Liver Transplantation Society. 2018.
6. Yilmaz S, Ceken K, Gurkan A, Erdogan O, Demirbas A, Kabaalioglu A, Sindel T, Luleci E. Endovascular treatment of a recipient celiac trunk stenosis after orthotopic liver transplantation. Journal of endovascular therapy : an official journal of the International Society of Endovascular Specialists. 2003;10(2):376-80.
7. Sharafuddin MJ, Olson CH, Sun S, Kresowik TF, Corson JD. Endovascular treatment of celiac and mesenteric arteries stenoses: applications and results. J Vasc Surg. 2003;38(4):692-8.